# SMARTSPANNER: Making SPANNER Robust in Low Resource Scenarios

**Min Zhang, Xiaosong Qiao, Yanqing Zhao, Shimin Tao, Hao Yang**

Huawei Translation Services Center, Beijing, China

{zhangmin186,qiaoxiaosong,zhaoyanqing,taoshimin,yanghao30}@huawei.com

## Abstract

Named Entity Recognition (NER) is one of the most fundamental tasks in natural language processing. Span-level prediction (SPANNER) is more naturally suitable for nested NER than sequence labeling (SEQLAB). However, according to our experiments, the SPANNER method is more sensitive to the amount of training data, i.e., the $F1$ score of SPANNER drops much more than that of SEQLAB when the amount of training data drops. In order to improve the robustness of SPANNER in low resource scenarios, we propose a simple and effective method SMARTSPANNER, which introduces a Named Entity Head (NEH[1]) prediction task to SPANNER and performs multi-task learning together with the task of span classification. Experimental results demonstrate that the robustness of SPANNER could be greatly improved by SMARTSPANNER in low resource scenarios constructed on the CoNLL03, FEW-NERD, GENIA and ACE05 standard benchmark datasets.

## 1 Introduction

NER is a fundamental information extraction task and plays an essential role in natural language processing applications such as information retrieval, question and answering, machine translation and knowledge graphs (Liu et al., 2022). The goal of NER is to extract named entities (NEs) into predefined categories, such as *person* (PER), *location* (LOC), *organization* (ORG) and *geo-political entity* (GPE). With the rapid evolution of neural architectures (Hochreiter and Schmidhuber, 1997; Kalchbrenner et al., 2014; Vaswani et al., 2017) and large pretrained models (Devlin et al., 2019; Brown et al., 2020; Lewis et al., 2020), recent years have seen the paradigm shift of NER systems from sequence labeling (Chiu and Nichols, 2016; Luo

---

[1]NEH is the first word of a named entity, for example, "Carl" is the NEH of "Carl Dinnon" in Fig. 1. If a named entity has only one word, the NEH is itself.

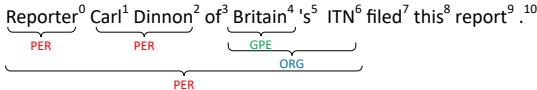

Figure 1: A sentence from ACE05 with 5 nested NEs. The superscript of each word indicates its index in the sentence.

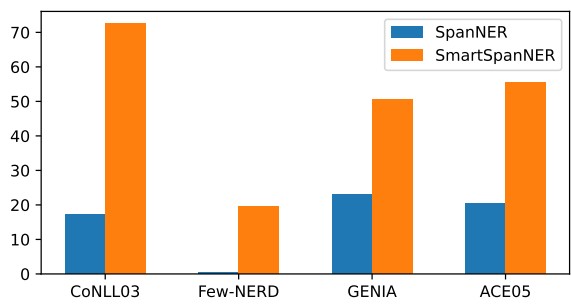

Figure 2: Averaged $F1$ scores of SPANNER and SMARTSPANNER methods over 10 independent runs on the test datasets of CoNLL03, FEW-NERD, GENIA and ACE05, where the models are trained on 1,000 sentences randomly sampled from the training data with the maximum epoch set to 10.

et al., 2020; Lin et al., 2020) to span prediction (Xu et al., 2017; Mengge et al., 2020; Tan et al., 2020; Fu et al., 2021). It is pointed out that nested entities are very common in the field of NER (Finkel and Manning, 2009), i.e., a named entity can contain or embed other entities as illustrated in Fig. 1. Compared with SEQLAB, SPANNER has an obvious advantage: All candidate entities can be easily found with different sub-sequences, which is straightforward for nested NER (Fu et al., 2021).

However, our experiments reveal that the performance of SPANNER drops much more than that of SEQLAB when the amount of training data drops, which makes SPANNER hard in low resource scenarios. In order to mitigate this problem, we propose a novel method SMARTSPANNER in

this paper. By introducing a Named Entity Head (NEH) prediction task for each word in given sentences, we perform multi-task learning together with the task of span classification for NER. We conduct experiments on both flat and nested standard benchmark datasets (CoNLL03, FEW-NERD, GENIA and ACE05). Experimental results demonstrate that SMARTSPANNER could improve the robustness in low resource scenarios significantly as shown in Fig 2.

Our contributions are summarized as follows:

- To the best of our knowledge, we are the first to highlight the robustness problem of SPANNER in low resource scenarios.

- To mitigate this problem, we develop a simple and effective method, named SMARTSPANNER. By introducing the task of NEH prediction, SMARTSPANNER can achieve significant gains in low resource scenarios.

- We provide an in-depth analysis of the reasons for the strong robustness of the method SMARTSPANNER.

## 2 Related Work

There has been a long history of research involving NER (McCallum and Li, 2003). Traditional approaches are based on Hidden Markov Model (HMM; Zhou and Su, 2002) or Conditional Random Field (CRF; Lafferty et al., 2001). With the development of deep learning technology (Hinton and Salakhutdinov, 2006), SEQLAB methods such as LSTM-CRF (Huang et al., 2015) and BERT-LSTM-CRF (Devlin et al., 2019) achieve very promising results in the field of NER. However, these methods cannot directly handle the nested structure because they can only assign one label to each token.

As it is pointed out that named entities are often nested (Finkel and Manning, 2009), various approaches for nested NER have been proposed in recent years (Wang et al., 2022; Shibuya and Hovy, 2020). One of the most representative directions is span-based methods that recognize nested entities by classifying sub-sequences of a sentence (Xu et al., 2017; Mengge et al., 2020; Tan et al., 2020; Fu et al., 2021; Zaratiana et al., 2022; Weng and Zhang, 2023). SPANNER methods are naturally suitable for the nested structure because nested entities can be easily detected in different sub-sequences. Although the strengths and weak-

| Percentage | 10% | 15% | 20% | 50% | 100% |
|---|---|---|---|---|---|
| # sentences | 1,498 | 2,248 | 2,997 | 7,493 | 14,987 |
| SEQLAB | 82.26 | 83.90 | 85.78 | 87.81 | 91.01 |
| SPANNER | 26.93 | 46.13 | 76.61 | 88.98 | 90.72 |

Table 1: $F1$ scores of SEQLAB and SPANNER methods on the test dataset of CoNLL03 with different percentages of training data and the maximum epoch set to 10.

nesses of SPANNER have been systematically investigated by Fu et al. (2021), its performance in low resource scenarios is not discussed. To the best of our knowledge, we are the first to highlight that the performance of SPANNER drops much more than that of SEQLAB when the amount of training data drops, which poses a challenge to the robustness of SPANNER in low resource scenarios. In order to address this challenge, we propose a novel method SMARTSPANNER.

It should be noted that the low resource scenarios in this paper refer to those with at least 1,000 labeled sentences, different from the settings for few-shot NER (Huang et al., 2021; Ding et al., 2021) and more common in real-world applications.

## 3 Methodology

### 3.1 Problem Description

The commonly-used NER standard benchmark dataset CoNLL03 (English) (Tjong Kim Sang and De Meulder, 2003) is selected to show the different sensitivities of SEQLAB and SPANNER methods.[2] 10%, 20%, 50% and 100% of the training data are used to train the models respectively, and the $F1$ scores on the test dataset are reported in Table 1.

From Table 1, although the performances of SEQLAB and SPANNER on the entire training data are almost the same, the $F1$ score of SPANNER drops much more than that of SEQLAB when the training data drops, i.e., the robustness of SPANNER in low resource scenarios needs to be greatly improved.

### 3.2 NEH for SPANNER

Given a sentence $S = \{w_1, \cdots, w_n\}$ with $n$ words, and a span $(i, j)$ denoting the sub-sequence in $S$ which starts with $w_i$ and ends with $w_j$, $w_i$ is an NEH if there is an NE span $(i, j)$. For example, the words "Reporter[0]", "Carl[1]" and "Britain[4]" in Fig. 1 are NEHs, and the other words are not.

From the above definition, we could conclude that $w_i$ is an NEH, which is a *necessary but not suf-*

---

[2]Details of implementation are described in Experiments.

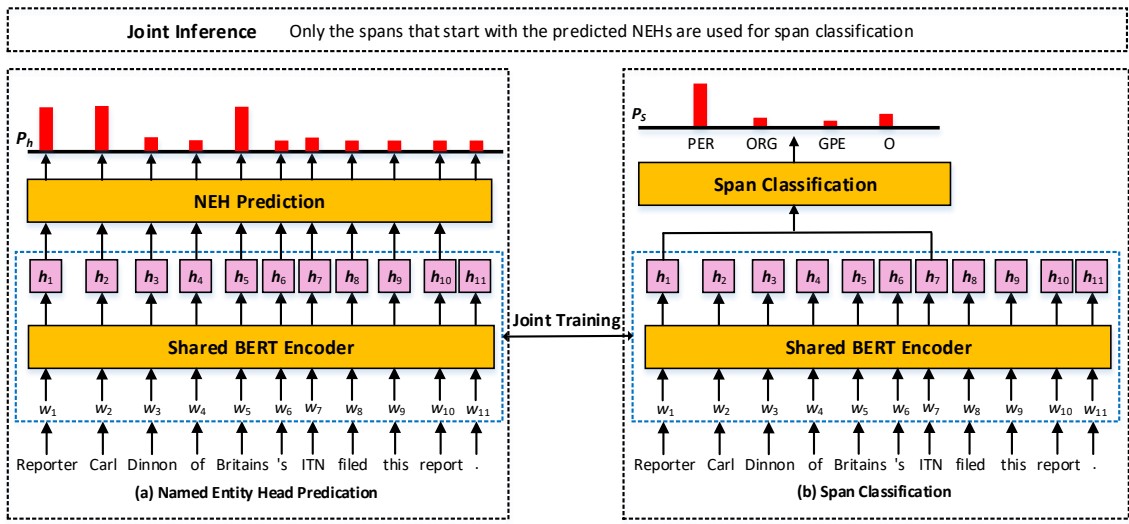

Figure 3: An overview of SMARTSPANNER method, which consists of (a) NEH prediction and (b) span classification. The two parts are jointly trained under the multi-task learning framework, and jointly determine the final results.

*ficient* condition for span $(i, j)$ to be an NE. That is to say, if $w_i$ is not an NEH, the span $(i, j)$ must not be an NE. Therefore, if we introduce the NEH prediction task for SPANNER, the number of spans for semantic tag classification could be greatly reduced in training and inferring. For example, without the NEH prediction task, the number of spans in Fig. 1 (all sub-sequences for the sentence with 11 words) for SPANNER is 66. If the three NEHs in Fig. 1 are correctly predicted, the number of spans (the sub-sequences starting with the words "Reporter[0]", "Carl[1]" and "Britain[4]") could be reduced to 28.

These are the basic ideas of our method SMARTSPANNER. With the introduction of the NEH prediction task, the number of spans for semantic tag classification could be greatly reduced, so the difficulty of semantic tag classification in SMARTSPANNER will be easier than in SPANNER. Meanwhile, the NEH prediction task is obviously easier than semantic tag classification (fewer categories and more balanced positive and negative samples). Therefore, with the help of the NEH prediction task, SMARTSPANNER reduces the difficulties of NER and could be more robust in low resource scenarios.

### 3.3 SMARTSPANNER

An overview of our SMARTSPANNER method is shown in Fig. 3, which consists of two parts: NEH prediction and span classification. NEH prediction aims to predict whether a word is the first word of an entity, and span classification aims to classify spans to corresponding semantic tags. The two parts are jointly trained under a multi-task learning framework with a shared encoder. The encoder is applied to a sentence to obtain contextual word representation, which is shared in downstream NEH prediction and span classification tasks. In this work, the NEH prediction task is treated as a special type of span classification, where the span width is 1 and the number of classes is 2. The span classification task is to aggregate the span information for multi-class classification. During inference, only the words predicted as NEHs are used to generate candidate spans for span classification.

#### 3.3.1 Encoder

Considering a sentence $S$ with $n$ tokenized words $\{w_i\}_{i=1}^n$, we convert the words to their contextual embeddings with BERT (Devlin et al., 2019) encoder. We generate the input sequence by concatenating a `[CLS]` token, $\{w_i\}_{i=1}^n$ and a `[SEP]` token, and use a series of $L$ stacked Transformer blocks (TBs) to project the input to a sequence of contextual vectors, i.e.,

$$\boldsymbol{h}_0, \cdots, \boldsymbol{h}_{n+1} = \mathrm{TB}_L([\texttt{CLS}], w_1, \cdots, w_n, [\texttt{SEP}]). \quad (1)$$

#### 3.3.2 Span Classification

Considering a span $(i, j)$, we use the boundary embeddings and span length embedding to represent

|  | CoNLL03 | | | FEW-NERD | | | GENIA | | | ACE05 | | |
|---|---|---|---|---|---|---|---|---|---|---|---|---|
|  | Train | Dev | Test | Train | Dev | Test | Train | Dev | Test | Train | Dev | Test |
| # sentences | 14,987 | 3,466 | 3,684 | 131,767 | 18,824 | 37,648 | 15,022 | 1,669 | 1,855 | 7,285 | 968 | 1,058 |
| # entities | 23,499 | 5,942 | 5,648 | 339,617 | 48,633 | 96,676 | 47,006 | 4,461 | 5,596 | 24,700 | 3,218 | 3,029 |
| # nested entities | - | - | - | - | - | - | 3,222 | 328 | 448 | 2,797 | 352 | 339 |
| avg length | 13.59 | 14.82 | 12.60 | 24.48 | 24.59 | 24.45 | 26.49 | 25.67 | 27.05 | 18.82 | 18.77 | 16.93 |

Table 2: Statistics of CoNLL03, FEW-NERD, GENIA and ACE05 datasets, where the number of entity types are 4, 66, 5 and 7 respectively.

the span (Fu et al., 2021):

$$\boldsymbol{s}(i,j) = [\boldsymbol{h}_i; \boldsymbol{h}_j; \boldsymbol{z}_{j-i+1}], \qquad (2)$$

where $\boldsymbol{z}_{j-i+1}$ is the span length embedding, which could be obtained by a learnable look-up table.

Next, we feed the span representation into a multi-layer perceptron (MLP) classifier, and apply a softmax layer to obtain the probability $P_s$ to predict its semantic tag.

$$P_s = \text{softmax}(\text{MLP}(\boldsymbol{s}(i,j))) \qquad (3)$$

Finally, we minimize the cross-entropy loss function:

$$\mathcal{L}_s = -\sum_{t=1}^{k}(y^t \log P_s^t + (1-y^t)\log(1-P_s^t)), \qquad (4)$$

where $k$ is the number of semantic tags, and $y^t$ denotes a label indicating whether the span $(i,j)$ is in tag $t$.

### 3.3.3 NEH Prediction

In this work, we treat NEH prediction as binary classification for a special span $(i,i)$ (i.e., only word $w_i$, $1 \leq i \leq n$) in sentence $S$. So the NEH probability $P_h$ is:

$$P_h = \text{softmax}(\text{MLP}(\boldsymbol{s}(i,i)))[:,1], \qquad (5)$$

and the cross-entropy loss function is:

$$\mathcal{L}_h = -(y_h \log P_h + (1-y_h)\log(1-P_h)), \qquad (6)$$

where $y_h$ denotes whether the word $w_i$ is an NEH.

### 3.3.4 Joint Training and Inference

We jointly minimize the following loss for training:

$$\mathcal{L} = w\mathcal{L}_h + (1-w)\mathcal{L}_s, \qquad (7)$$

where $\mathcal{L}_h$ and $\mathcal{L}_s$ are the losses of the NEH prediction task and span classification task, and $w$ is the hyper-parameter to balance the two tasks.

During inference, only the words predicted as NEHs are used to generate candidate spans for span classification.

Therefore, in order to keep the training and inferring data distributions consistent, training the span classification model in SMARTSPANNER only requires a part of the spans that are all needed in SPANNER. A selection strategy is designed for each span $(i,j)$ in training:

$$\text{Select}((i,j)) = \begin{cases} 1, & \text{if } w_i \text{ is an NEH} \\ 1, & \text{if } rand < sp \\ 0, & \text{otherwise} \end{cases}, \qquad (8)$$

where $rand$ is a randomly generated float number in $[0, 1]$ and $sp$ is a hyper-parameter of the selection threshold (0.05 used in this paper).

An example for this selection strategy is provided to compare the number of total training samples in SPANNER and SMARTSPANNER: Given a sentence having $n$ tokens and $d$ named entities, we could get the number of training samples generated by this sentence for SPANNER ($N_{sn}$) and SMARTSPANNER ($N_{ssn}$) respectively when the max span width is set to the value $m$:

$$\begin{aligned} N_{sn} &= n * m - (m-1) * m/2 \\ N_{ssn} &\approx n + d * m + (n-d) * m * sp \end{aligned} \qquad (9)$$

It should be noted that the training samples of NEH prediction are included in $N_{ssn}$ (the first term $n$ in the right). Supposing $n = 100, d = 5, m = 10, sp = 0.05$, we could have $N_{sn} = 955, N_{ssn} \approx 197$. This means the training samples are greatly reduced in SMARTSPANNER, and thus the training time is also greatly reduced. This is the reason why we name the method "SMART".

## 4 Experiments

### 4.1 Datasets

Four standard benchmark datasets CoNLL03 English (Tjong Kim Sang and De Meulder, 2003),

| # Sents | Methods | CoNLL03 | | | FEW-NERD | | | GENIA | | | ACE05 | | |
|---------|---------|-------|-------|-------|-------|-------|-------|-------|-------|-------|-------|-------|-------|
| | | $P$ | $R$ | $F1$ | $P$ | $R$ | $F1$ | $P$ | $R$ | $F1$ | $P$ | $R$ | $F1$ |
| 1,000 | SPANNER | 45.33 | 10.92 | 17.28 | 0.00 | 0.00 | 0.00 | 42.73 | 17.28 | 23.06 | 66.58 | 12.46 | 20.44 |
| | SMARTSPANNER | 77.85 | 67.75 | 72.45 | 49.74 | 12.19 | **19.57** | 41.36 | 65.46 | **50.66** | 51.03 | 60.60 | **55.38** |
| | SEQLAB | 75.24 | 82.52 | **78.38** | 14.47 | 20.99 | 17.10 | - | - | - | - | - | - |
| 2,000 | SPANNER | 73.98 | 59.11 | 65.69 | 21.85 | 0.00 | 0.01 | 38.26 | 45.67 | 41.54 | 53.92 | 45.02 | 48.98 |
| | SMARTSPANNER | 87.16 | 80.66 | **83.78** | 48.54 | 27.02 | **34.69** | 54.50 | 77.35 | **63.94** | 62.64 | 76.37 | **68.82** |
| | SEQLAB | 80.85 | 86.37 | 83.52 | 27.08 | 35.31 | 30.71 | - | - | - | - | - | - |
| 5,000 | SPANNER | 87.24 | 84.98 | 86.09 | 55.39 | 19.29 | 28.62 | 57.00 | 73.72 | 64.28 | 63.99 | 77.62 | 70.14 |
| | SMARTSPANNER | 89.98 | 89.88 | **89.93** | 57.97 | 58.56 | **58.26** | 65.67 | 81.20 | **72.61** | 70.49 | 84.33 | **76.79** |
| | SEQLAB | 85.07 | 89.20 | 87.08 | 44.64 | 53.11 | 48.50 | - | - | - | - | - | - |
| ALL | SPANNER | 90.15 | 91.29 | 90.72 | 67.62 | 70.99 | **69.26** | 70.26 | 81.94 | 75.65 | 73.11 | 85.83 | 78.96 |
| | SMARTSPANNER | 91.25 | 91.56 | **91.40** | 67.45 | 70.23 | 68.81 | 71.61 | 82.22 | **76.54** | 77.51 | 87.05 | **82.00** |
| | SEQLAB | 89.87 | 92.19 | 91.01 | 67.05 | 69.40 | 68.20 | - | - | - | - | - | - |

Table 3: Overall results of SPANNER, SMARTSPANNER and SEQLAB on CoNLL03, FEW-NERD, GENIA and ACE05. The $P$, $R$ and $F1$ are the mean values in 10 independent runs. Best $F1$ scores are in bold. "# Sents" stands for the number of sentences used for training.

FEW-NERD (SUP) (Ding et al., 2021), GENIA (Kim et al., 2003) and ACE05[3] English are selected for evaluation, where CoNLL03 and FEW-NERD are for flat NER and the others are for nested NER. The statistics of these datasets are shown in Table 2. It should be pointed out that we follow Shibuya and Hovy (2020)'s preprocessing steps[4] to split GENIA and ACE05 into train, development, and test sets.

## 4.2 Experiment Settings

In experiments, we implement the SPANNER and SMARTSPANNER methods based on the source codes[5] by Fu et al. (2021), where the pretrained model `bert-base-uncased`[6] is used as the encoder. And we implement BERT-CRF as the SEQLAB method for comparison.

The training datasets for low resource scenarios are constructed by `random.shuffle` on the entire training sentences and extracting the first 1,000, 2,000 or 5,000 sentences, where the random seeds in 10 runs are set to 1, 2, 3, 4, 5, 6, 7, 8, 9 and 42 respectively. In addition, the random seed for the results in Table 1 is set to 42. The development datasets of CoNLL03, GENIA and ACE05 are all used in the constructed low resource scenarios. As the development data in FEW-NERD is very large (18,824 sentences), we choose the first 2,000 sentences as the development data when the training data contains 1,000, 2,000 or 5,000 sentences.

For the SPANNER and SMARTSPANNER methods, the embedding size of the span width is set

to 50, the max span width is set to 20[7]. The MLP for span classification takes two layers, which is the same as the setting by Fu et al. (2021). Considering that NEH classification is less challenging compared to span classification, we use single-layer MLP for NEH classification.

The training epoch number is set to 10, and the batch size is set to 16. During training, the SPANNER, SMARTSPANNER and SEQLAB models are optimized by AdamW (Loshchilov and Hutter, 2019) with the learning rate set to 0.00001 and a linear warmup scheduler. The values of the hyper-parameters $w$ for joint training and $sp$ for span selection in SMARTSPANNER are set to 0.2 and 0.05 respectively. All models are trained using a single NVIDIA V100 GPU.

## 4.3 Main Results

The results of SPANNER, SMARTSPANNER and SEQLAB on the test datasets of CoNLL03, FEW-NERD, GENIA and ACE05 are reported in Table 3, where the values of the precision ($P$), recall ($R$) and $F1$ score are included (SEQLAB only for flat NER datasets, not for nested).

From the results of SPANNER and SMARTSPANNER on 1,000, 2,000, and 5,000 training sentences, it is obvious that the $F1$ scores of SMARTSPANNER are greatly better that those of SPANNER on all the four datasets. Specially, when the training data is 1,000 sentences, compared with SPANNER, SMARTSPANNER has the most obviously improvement in $F1$ scores (17.28% to 72.45% on CoNLL03, 0.00% to 19.57% on FEW-NERD,

[3] catalog.ldc.upenn.edu/LDC2006T06
[4] https://github.com/yahshibu/nested-ner-tacl2020
[5] https://github.com/neulab/SpanNER
[6] https://huggingface.co/bert-base-uncased

[7]This setting is to ensure the proportions of entities with a length exceeding 20 in the four datasets do not exceed 1%.

| # Sents | Methods | CoNLL03 | FEW-NERD | GENIA | ACE05 |
|---|---|---|---|---|---|
| 1,000 | SN | 6.9 | 10.0 | 10.5 | 8.0 |
| | SSN | 4.9 | 6.4 | 7.1 | 5.7 |
| | | ↓29% | ↓36% | ↓33% | ↓29% |
| 2,000 | SN | 13.7 | 19.8 | 21.2 | 15.9 |
| | SSN | 9.9 | 12.8 | 14.2 | 11.3 |
| | | ↓28% | ↓36% | ↓33% | ↓29% |
| 5,000 | SN | 33.6 | 49.5 | 53.7 | 35.1 |
| | SSN | 24.4 | 31.9 | 35.7 | 26.0 |
| | | ↓27% | ↓35% | ↓34% | ↓26% |
| ALL | SN | 100.1 | 1311.3 | 162.4 | 58.3 |
| | SSN | 72.6 | 844.8 | 108.0 | 41.2 |
| | | ↓27% | ↓36% | ↓33% | ↓29% |

Table 4: Training time (in seconds) of SPANNER (SN) and SMARTSPANNER (SNN) methods on the datasets of CoNLL03, FEW-NERD, GENIA and ACE05.

| # Sents | Methods | CoNLL03 | FEW-NERD | GENIA | ACE05 |
|---|---|---|---|---|---|
| 1,000 | SN | 40.5 | 1029.2 | 37.6 | 12.7 |
| | SSN | 14.7 | 301.7 | 12.2 | 5.4 |
| | | ↓64% | ↓71% | ↓68% | ↓57% |
| 2,000 | SN | 40.7 | 1026.7 | 37.9 | 12.7 |
| | SSN | 14.0 | 280.3 | 11.8 | 5.1 |
| | | ↓66% | ↓73% | ↓69% | ↓60% |
| 5,000 | SN | 41.0 | 1041.0 | 37.8 | 12.7 |
| | SSN | 13.5 | 277.3 | 11.5 | 5.2 |
| | | ↓67% | ↓73% | ↓70% | ↓59% |
| ALL | SN | 41.1 | 1041.7 | 37.8 | 12.7 |
| | SSN | 13.5 | 270.1 | 11.1 | 5.2 |
| | | ↓67% | ↓74% | ↓71% | ↓59% |

Table 5: Inferring time (in seconds) of SPANNER (SN) and SMARTSPANNER (SNN) methods on the test datasets of CoNLL03, FEW-NERD, GENIA and ACE05.

23.06% to 50.66% on GENIA, and 20.44% to 55.38% on ACE05). Therefore, SMARTSPANNER is much more robust in low resource scenarios.

From the comparison results of SEQLAB and SMARTSPANNER on the two flat NER datasets, it can be seen that the $F1$ scores of SMARTSPANNER are better than those of SEQLAB on all low resource settings except on the 1,000 sentences of CoNLL03. It is worth mentioning that SMARTSPANNER is more effective than SEQLAB on all the low resource settings of FEW-NERD — using such settings poses a significant challenging due to the large number of entity types in FEW-NERD.

Therefore, by introducing the NEH prediction task, SMARTSPANNER greatly improves the robustness of SPANNER in low resource scenarios, even better than SEQLAB.

## 4.4 Efficiency

According to our analysis, compared with SPANNER, SMARTSPANNER reduces the number of samples (spans) for training and inferring. To verify this, we compare the efficiencies of SPANNER and SMARTSPANNER in this section. Table 4 and

| # Sents | Methods | Tasks | # CAT | # PS | # NS | # PS / # NS |
|---|---|---|---|---|---|---|
| 1,000 | SSN | NEH | 2 | 3,072 | 15,553 | 1/5.1 |
| | | SP | 8 | 3,341 | 47,168 | 1/13.1 |
| | SN | SP | 8 | 3,341 | 207,281 | 1/62.0 |
| 2,000 | SSN | NEH | 2 | 6,100 | 30,874 | 1/5.1 |
| | | SP | 8 | 6,668 | 87,177 | 1/13.1 |
| | SN | SP | 8 | 6,668 | 412,529 | 1/61.9 |
| 5,000 | SSN | NEH | 2 | 15,584 | 78,179 | 1/5.0 |
| | | SP | 8 | 16,973 | 222,745 | 1/13.1 |
| | SN | SP | 8 | 16,973 | 1,053,331 | 1/62.1 |
| ALL | SSN | NEH | 2 | 22,614 | 114,524 | 1/5.1 |
| | | SP | 8 | 24,614* | 323,921 | 1/13.2 |
| | SN | SP | 8 | 24,614* | 1,545,836 | 1/62.8 |

* The value is 24,614, not 24,700 shown in Table 2. This is because spans exceeding the max span width (set to 20 in experiments) are not used for training.

Table 6: Descriptions of ACE05's training data for the tasks in SMARTSPANNER (SSN) and SPANNER (SN) methods, where # CAT, # PS and # NS mean the number of classification categories, the number of positive samples and the number of negative samples respectively.

Table 5 report the running time of the two methods on the four datasets (CoNLL03, FEW-NERD, GENIA and ACE05) during training and inferring. For training, the running time is the average time to train one epoch. For inferring, the running time is the time to complete NER on the test datasets.

From Table 4, it could be seen that SMARTSPANNER takes at least 26% less training time than SPANNER on both all low resource settings and the entire data of all the four datasets. This is because much fewer negative samples for span classification are used in the training of SMARTSPANNER, according to the selection strategy.

From Table 5, it could be seen that the inferring time of SMARTSPANNER is at least 57% shorter than that of SPANNER on the four test datasets of CoNLL03, FEW-NERD, GENIA and ACE05. This is because only the spans that start with the predicted NEHs are used for span classification in SMARTSPANNER, which reduces the number of spans for inferring greatly.

From the comparison results in Table 4 and Table 5, SMARTSPANNER is much more efficient than SPANNER during training and inferring on all low resource settings and the entire data of all the four datasets.

## 5 Analysis

We have shown the robustness of SMARTSPANNER in low resource scenarios. In this section, we aim to take a deeper look and understand what contributes to its final performance.

| Methods | Tasks | # Sentences | | | |
|---|---|---|---|---|---|
| | | 1,000 | 2,000 | 5,000 | ALL |
| SSN | NEH | 17,909 | 17,909 | 17,909 | 17,909 |
| | SP | 40,743 | 38,409 | 37,874 | 37,553 |
| SN | SP | 192,302 | 192,302 | 192,302 | 192,302 |

Table 7: Number of inferring samples for the tasks in SMARTSPANNER (SSN) and SPANNER (SN) methods on the test dataset of ACE05 (1,000, 2,000, 5,000 and all sentences for training respectively).

## 5.1 Task Analysis

SMARTSPANNER has two tasks, i.e., NEH prediction task and span classification (SP) task, while SPANNER has one task, i.e., SP task. We first provide a detailed comparison of training data for these tasks on ACE05, which are shown in Table 6.

From Table 6, it could be found that the task of NEH prediction is the easiest, because the number of classification categories is the smallest and the balance of positive and negative samples[8] is the best. Meanwhile, due to the more balanced positive and negative samples brought by the selection strategy, the task of span classification in SMARTSPANNER is much easier than that in SPANNER. Although deep learning methods can solve difficult problems, they require large amounts of data. Therefore, SMARTSPANNER is more effective than SPANNER for NER in low-resource scenarios. Furthermore, it could be seen that the total samples of SMARTSPANNER for training are much less than those of SPANNER (about 70% reduction). This is why the training time of SMARTSPANNER is much less than that of SPANNER.

Next, we compare the number of inferring samples for the tasks in SMARTSPANNER and SPANNER on the test dataset of ACE05 when 1,000, 2,000, 5,000 and all sentences are used for training, as shown in Table 7. It could be clearly seen that the total inferring samples in SMARTSPANNER are greatly less than those in SPANNER (more than 70% reduction). This is why the inferring time of SMARTSPANNER is much less.

Finally, we provide the results of the two tasks in SMARTSPANNER on the test dataset of ACE05 when 1,000, 2,000, 5,000 and all sentences are used for training data. In Table 8, the rows of "NEH" list the results of the NEH prediction task, the rows of "SP" list the results of the SP task on all the possible spans, and the rows of "NEH + SP" list

---

[8]The positive samples means the spans are NEHs in NEH prediction task or NEs in span classification task, otherwise they are negative samples.

| # Sents | Tasks | $P$ | $R$ | $F1$ |
|---|---|---|---|---|
| 1,000 | NEH | 78.03 | 91.84 | 84.33 |
| | SP | 39.33 | 62.12 | 48.13 |
| | NEH + SP | 51.03 | 60.60 | 55.38 (↑ **7.25**) |
| 2,000 | NEH | 84.44 | 94.86 | 89.34 |
| | SP | 48.60 | 77.98 | 59.87 |
| | NEH + SP | 62.64 | 76.37 | 68.82 (↑ **8.95**) |
| 5,000 | NEH | 87.66 | 95.91 | 91.60 |
| | SP | 60.68 | 85.92 | 71.12 |
| | NEH + SP | 70.49 | 84.33 | 76.79 (↑ **5.67**) |
| ALL | NEH | 89.76 | 95.91 | 92.73 |
| | SP | 65.04 | 88.51 | 73.71 |
| | NEH + SP | 77.51 | 87.05 | 82.00 (↑ **8.29**) |

Table 8: The average values of precision ($P$), recall ($R$) and $F1$ scores of the two tasks (NEH and SP) in SMARTSPANNER on the test dataset of ACE05 in 10 independent runs (1,000, 2,000, 5,000 and all sentences for training respectively).

the results of SMARTSPANNER (only the spans that start with the predicted NEHs are used for span classification).

From Table 8, it could be seen that the $F1$ scores of NEH prediction tasks are higher than those of span classification tasks due to the lower task difficulty. When span classification in SMARTSPANNER is used for all possible spans, the precision suffers because of the inconsistent distributions of training and test data (worse than that of SPANNER in Table 3). When NEH prediction is used before span classification (i.e., only the spans that start with the predicted NEHs are used for span classification), the precision rates are greatly improved (more than 10%), the recall rates are slightly decreased (less than 2%), and the $F1$ scores are significantly improved (more than 5%). The reason for the drop of recall rates is that the recall rates of NEH are not 100%. Two cases from the test dataset of ACE05 are shown in A.1.

The analyses on CoNLL03, FEW-NERD and GENIA are provided in A.2.

## 5.2 Hyper-Parameter Analysis

There are two hyper-parameters (sample selection probability $sp$ and joint training weight $w$) for training SMARTSPANNER. In this section, we provide a detailed analysis for the values of these two hyper-parameters respectively.

### 5.2.1 Sample Selection Probability

We vary the sample selection probability parameter $sp$ from 0 to 1 with step 0.05, and perform 10 independent runs of SMARTSPANNER for each

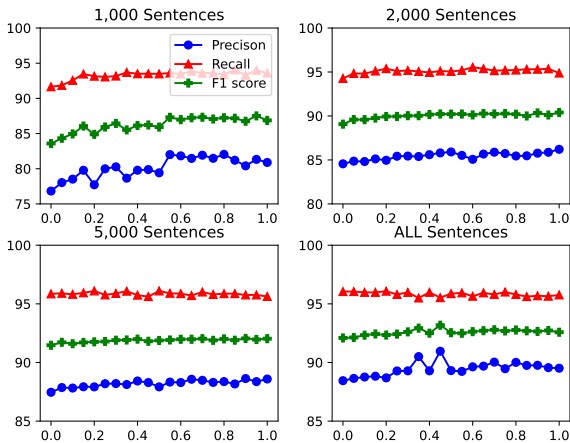

Figure 4: The average values of Precision, Recall and $F1$ scores of NEH prediction on the test dataset of ACE05 with different sample selection probabilities $sp$ (from 0 to 1 with step 0.05) in 10 independent runs when 1,000, 2,000, 5,000 and all sentences are used for training respectively.

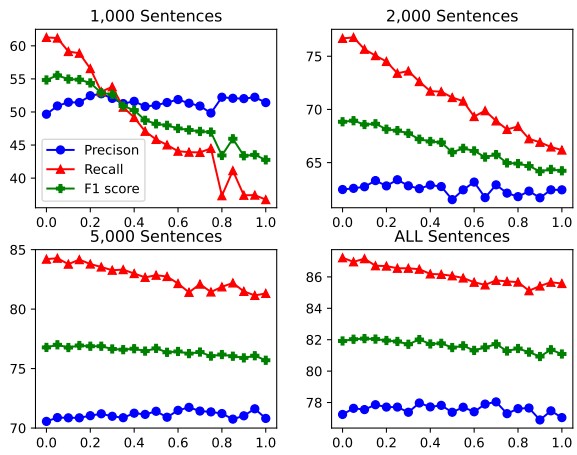

Figure 5: The average values of Precision, Recall and $F1$ scores of SMARTSPANNER on the test dataset of ACE05 with different sample selection probabilities $sp$ (from 0 to 1 with step 0.05) in 10 independent runs when 1,000, 2,000, 5,000 and all sentences are used for training respectively.

value of $sp$ on the dataset ACE05. Fig. 4 and Fig. 5 show the results (Precision, Recall and $F1$ scores) of NEH prediction and NER in SMARTSPANNER respectively when the parameter $sp$ varies (the joint training weight $w$ remains at 0.2).

From Fig. 4, it could be seen that the recall rates of NEH prediction hardly change and the precision rates improve slightly with the increase of the parameter $sp$. Therefore, the change of the parameter $sp$ has little effect on the task of NEH prediction. That is to say, despite the parameter $sp$ changes, the distribution of the data for span classification

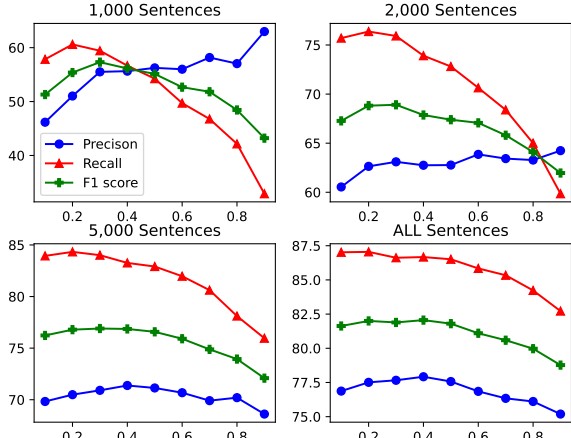

Figure 6: The average values of Precision, Recall and $F1$ scores of SMARTSPANNER on the test dataset of ACE05 with different joint training weight $w$ (from 0.1 to 0.9 with step 0.1) in 10 independent runs when 1,000, 2,000, 5,000 and all sentences are used for training respectively.

in SMARTSPANNER during inference is stable.

From Fig. 5, it could be seen that the $F1$ scores of NER in SMARTSPANNER decreases with the increase of the parameter $sp$, especially when the training data is small (such as 1,000 sentences). This is due to the inconsistency between training data and inferring data for span classification in SMARTSPANNER (the larger the parameter $sp$, the higher the inconsistency). Since the robustness of deep learning methods will improve as the amount of data increases, SMARTSPANNER is less sensitive to the parameter $sp$ when the training data increases (such as all sentences). From Fig. 5, it could be seen that 0.05 is a good choice for the parameter $sp$ in low resource scenarios.

### 5.2.2 Joint Training Weight

We vary the joint training weight $w$ from 0.1 to 0.9 with step 0.1 and perform 10 independent runs of SMARTSPANNER for each value of $w$ on the dataset ACE05. Fig. 6 shows the results (Precision, Recall and $F1$ scores) of NER when the parameter $w$ varies (the sample selection probability parameter $sp$ remains at 0.05).

From Fig. 6, it could be found that the value of parameter $w$ between 0.2 and 0.4 is suitable for SMARTSPANNER according to the $F1$ scores. And the sensitivity of SMARTSPANNER to parameter $w$ decreases with the increase of training data.

# 6 Conclusion

In this paper, it is found that the SPANNER method is sensitive to the amount of training data, i.e., the performance of SPANNER is worse than SEQLAB in low resource scenarios. In order to alleviate this problem, SMARTSPANNER is proposed by introducing the NEH prediction task into SPANNER in a multi-task learning manner. The comparison results of experiments designed on the CoNLL03, FEW-NERD, GENIA and ACE05 datasets demonstrate that SMARTSPANNER is much more robust in low resource scenarios than SPANNER, and greatly reduces the running time of training and inferring. In addition, the reasons for the strong robustness of SMARTSPANNER are analyzed in depth on the dataset ACE05.

## Limitations

The SMARTSPANNER method proposed in this paper is very effective in low resource scenarios. However, when the training data contains more than 10,000 sentences, compared with SPANNER, the advantages of SMARTSPANNER on the CoNLL03 and GENIA datasets are not so obvious. Furthermore, when all the training data of the FEW-NERD dataset (more than 100,000 sentences) is used, the results of SMARTSPANNER even drop a bit. Therefore, SMARTSPANNER is not strongly recommended in high resource scenarios.

## Ethics Statement

In this section, we discuss the ethical consideration of this work from the following two aspects. First, for SMARTSPANNER, the code, data and pretrained models adopted from previous works are granted for research-purpose usage. Second, SMARTSPANNER improves the robustness of SPANNER in low resource scenarios by introducing the NEH prediction task. Hence we do not foresee any major risks or negative societal impact of our work. However, like any other ML models, the named entities recognized by our model may not always be completely accurate and hence should be used with caution for real-world applications.

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

## A    Appendix

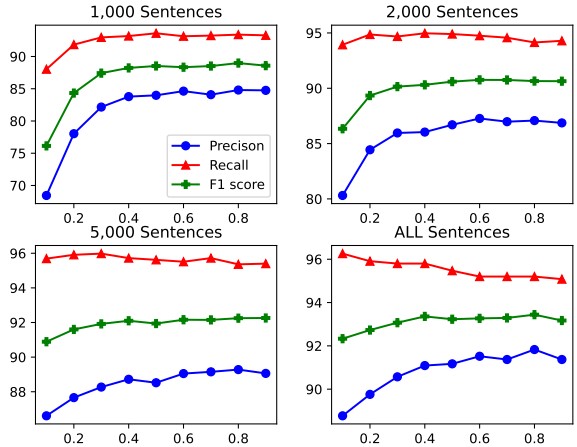

Figure 7: The average values of Precision, Recall and $F1$ scores of NEH prediction in SMARTSPANNER on the test dataset of ACE05 with different joint training weight $w$ (from 0.1 to 0.9 with step 0.1) in 10 independent runs when 1,000, 2,000, 5,000 and all sentences are used for training respectively.

| Dataset | SPANNER | | | SMARTSPANNER | | |
|---|---|---|---|---|---|---|
| | $P$ | $R$ | $F1$ | $P$ | $R$ | $F1$ |
| CoNLL03 | 74.25 | 56.11 | 63.67 | 88.98 | 88.87 | **88.92** |
| FEW-NERD | 0.00 | 0.00 | 0.00 | 47.81 | 29.90 | **36.67** |
| GENIA | 50.09 | 30.64 | 37.81 | 61.72 | 71.61 | **66.28** |
| ACE05 | 72.41 | 12.05 | 19.60 | 70.29 | 76.44 | **73.23** |

Table 9: Averaged $P$, $R$ and $F1$ values of SPANNER and SMARTSPANNER methods over 3 independent runs, where the models are trained on 1,000 sentences randomly sampled from the training data with the maximum epoch set to 25.

### A.1    Case Study

Two cases from ACE05 are shown in Table 10, where the results of SPANNER and SMARTSPANNER are provided. From Table 10, it could be clearly seen that SMARTSPANNER performs better for NER than SPANNER, and the predicted NEHs are beneficial to improve the precision of NER in SMARTSPANNER.

### A.2    Supplement Experimental Results

We provide the results of task analysis on the datasets CoNLL03, FEW-NERD and GENIA,

which are shown in Table 11, Table 12 and Table 13. From Table 11, it could be seen that SMARTSPANNER also obtains much more balanced positive and negative samples on the three datasets. As shown in Table 11 and Table 12, SMARTSPANNER requires significantly fewer samples for training and inferring on the three datasets than SPANNER, enabling much faster training and inferring. Table 13 demonstrates the results of the two tasks in SMARTSPANNER on the three datasets, which are consistent with the results on ACE05.

For reference, we report the results of NEH prediction in SMARTSPANNER on ACE05 with different joint training weight $w$ in Fig. 7.

According to the results (Zaratiana et al., 2022), setting a larger epoch value will lead to better results for SpanNER. Therefore, we just increase the training epoch number from 10 to 25 and conduct three independent experiments on the four datasets (using random seeds 1, 2 and 42 to obtain 1,000 training sentences). The comparison results are reported in Table 9. It can be seen that setting a larger epoch number only results in a significant improvement of $F1$ score for SPANNER on CoNLL03 (with minor improvements on GENIA and ACE05, and no change on FEW-NERD). Moreover, the superiority of SMARTSPANNER over SPANNER remains evident across all four datasets, highlighting the robustness of SMARTSPANNER in low-resource scenarios.

| | | |
|---|---|---|
| **Case 1** | **Sentence** | He[0] was[1] the[2] governor[3] of[4] my[5] state[6] of[7] Texas[8] ,[9] where[10] there[11] are[12] a[13] whole[14] lot[15] of[16] doctors[17] .[18] |
| | **Ground Truth NEs** (Total 7) | (0, 0, PER), (2, 17, PER), (5, 5, PER), (5, 17, GPE), (8, 8, GPE), (10, 10, GPE), (13, 17, PER) |
| | **Ground Truth NEHs** (Total 6) | (0, 0), (2, 2), (5, 5), (8, 8), (10, 10), (13, 13) |
| | **NEs by SPANNER** (2 correct, 2 incorrect) | (0, 0, PER, 0.8446), (5, 8, GPE, 0.4203), (5, 17, PER, 0.4736), (8, 8, GPE, 0.6953) |
| | **NEHs by SMARTSPANNER** (6 correct, 1 incorrect) | (0, 0, 0.9961), (2, 2, 0.9769), (5, 5, 0.9995), (8, 8, 0.9005), (10, 10, 0.9729), (11, 11, 0.7172), (13, 13, 0.9433) |
| | **NEs by SMARTSPANNER** (5 correct, 1 incorrect) | (0, 0, PER, 0.9465), (5, 5, PER, 0.8316), (8, 8, GPE, 0.9542), (10, 10, GPE, 0.7904), (11, 17, PER, 0.6768), (13, 17, PER, 0.8246) |
| | **NEs by SMARTSPANNER without NEHs** (5 correct, 3 incorrect) | (0, 0, PER, 0.9465), (5, 5, PER, 0.8316), (6, 8, GPE, 0.6781), (8, 8, GPE, 0.9542), (10, 10, GPE, 0.7904), (11, 17, PER, 0.6768), (13, 17, PER, 0.8246), (17, 17, PER, 0.7493) |
| **Case 2** | **Sentence** | A[0] traveler[1] was[2] driving[3] through[4] Arkansas[5] when[6] he[7] lost[8] his[9] way[10] and[11] got[12] off[13] the[14] main[15] highway[16] .[17] |
| | **Ground Truth NEs** (Total 5) | (0, 1, PER), (5, 5, GPE), (7, 7, PER), (9, 9, PER), (14, 16, FAC) |
| | **Ground Truth NEHs** (Total 5) | (0, 0), (5, 5), (7, 7), (9, 9), (14, 14) |
| | **NEs by SPANNER** (4 correct, 1 incorrect) | (0, 1, PER, 0.8230), (0, 7, PER, 0.7019), (5, 5, GPE, 0.7611), (7, 7, PER, 0.8407), (9, 9, PER, 0.5256) |
| | **NEHs by SMARTSPANNER** (5 correct, 0 incorrect) | (0, 0, 0.9886), (5, 5, 0.9680), (7, 7, 0.9959), (9, 9, 0.9899), (14, 14, 0.9180) |
| | **NEs by SMARTSPANNER** (5 correct, 1 incorrect) | (0, 1, PER, 0.9403), (0, 5, PER, 0.5006), (5, 5, GPE, 0.9541), (7, 7, PER, 0.8839), (9, 9, PER, 0.7767), (14, 16, FAC, 0.4102) |
| | **NEs by SMARTSPANNER without NEHs** (5 correct, 2 incorrect) | (0, 1, PER, 0.9403), (0, 5, PER, 0.5006), (1, 1, PER, 0.6204), (5, 5, GPE, 0.9541), (7, 7, PER, 0.8839), (9, 9, PER, 0.7767), (14, 16, FAC, 0.4102) |

Table 10: Two cases from ACE05. The superscript of each word indicates its index in the sentence, the ground truth NEs are represented by triples (NE start word index, NE end word index, NE type), and the ground truth NEHs are represented by special spans (same start and end indexes). The NEs predicted by SPANNER and SMARTSPANNER are represented by four-tuples (predicted NE start word index, predicted NE end word index, predicted NE type, predicted probability), and the NEHs predicted by SMARTSPANNER are represented by triples (predicted start index, predicted end index, predicted probability). The triples or four-tuples in red indicate incorrect predictions. The rows of "NEs by SMARTSPANNER without NEHs" show the predicted NEs by SMARTSPANNER when all possible spans are used for NE prediction (i.e., the predicted NEHs are not used for span filtering).

| # Sents | Methods | Tasks | CoNLL03 | | | | FEW-NERD | | | | GENIA | | | |
|---|---|---|---|---|---|---|---|---|---|---|---|---|---|---|
| | | | # CAT | # PS | # NS | # PS / # NS | # CAT | # PS | # NS | # PS / # NS | # CAT | # PS | # NS | # PS / # NS |
| 1,000 | SSN | NEH | 2 | 1,620 | 12,446 | 1/7.7 | 2 | 2,598 | 22,190 | 1/8.5 | 2 | 2,859 | 23,223 | 1/8.1 |
| | | SP | 5 | 1,620 | 22,636 | 1/14.0 | 67 | 2,598 | 46,351 | 1/17.8 | 6 | 3,018 | 49,478 | 1/16.4 |
| | SN | SP | 5 | 1,620 | 150,140 | 1/92.7 | 67 | 2,598 | 311,226 | 1/119.8 | 6 | 3,018 | 333,423 | 1/110.5 |
| 2,000 | SSN | NEH | 2 | 3,244 | 24,794 | 1/7.6 | 2 | 5,214 | 44,146 | 1/8.5 | 2 | 5,705 | 46,968 | 1/8.2 |
| | | SP | 5 | 3,244 | 44,843 | 1/13.8 | 67 | 5,214 | 92,903 | 1/17.8 | 6 | 6,054 | 99,718 | 1/16.5 |
| | SN | SP | 5 | 3,244 | 298,936 | 1/92.2 | 67 | 5,214 | 619,037 | 1/118.7 | 6 | 6,054 | 677,073 | 1/111.8 |
| 5,000 | SSN | NEH | 2 | 7,862 | 61,086 | 1/7.8 | 2 | 12,942 | 109,878 | 1/8.5 | 2 | 14,513 | 117,807 | 1/8.1 |
| | | SP | 5 | 7,862 | 107,335 | 1/13.7 | 67 | 12,942 | 229,465 | 1/17.7 | 6 | 15,423 | 254,543 | 1/16.5 |
| | SN | SP | 5 | 7,862 | 727,192 | 1/92.5 | 67 | 12,942 | 1,538,134 | 1/118.8 | 6 | 15,423 | 1,705,363 | 1/110.6 |
| ALL | SSN | NEH | 2 | 23,499 | 181,068 | 1/7.7 | 2 | 339,602 | 2,880,060 | 1/8.5 | 2 | 43,536 | 354,703 | 1/8.1 |
| | | SP | 5 | 23,499 | 319,074 | 1/13.6 | 67 | 339,602 | 6,013,526 | 1/17.7 | 6 | 46,984 | 766,669 | 1/16.3 |
| | SN | SP | 5 | 23,499 | 2,146,032 | 1/91.3 | 67 | 339,602 | 40,291,745 | 1/118.6 | 6 | 46,984 | 5,133,236 | 1/109.3 |

Table 11: Descriptions of the training data of CoNLL03, FEW-NERD and GENIA for the tasks in SMARTSPANNER (SSN) and SPANNER (SN) methods, where # CAT means the number of classification categories, # PS means the number of positive samples, and # NS means the number of negative samples.

| Methods | Tasks | CoNLL03 (# Sentences) | | | | FEW-NERD (# Sentences) | | | | GENIA (# Sentences) | | | |
|---|---|---|---|---|---|---|---|---|---|---|---|---|---|
| | | 1,000 | 2,000 | 5,000 | ALL | 1,000 | 2,000 | 5,000 | ALL | 1,000 | 2,000 | 5,000 | ALL |
| SSN | NEH | 46,666 | 46,666 | 46,666 | 46,666 | 919,162 | 919,162 | 919,162 | 919,162 | 46,878 | 46,878 | 46,878 | 46,878 |
| | SP | 69,758 | 62,204 | 60,424 | 60,323 | 1,697,569 | 1,581,779 | 1,519,050 | 1,452,749 | 93,219 | 85,390 | 80,286 | 78,402 |
| SN | SP | 477,887 | 477,887 | 477,887 | 477,887 | 11,595,325 | 11,595,325 | 11,595,325 | 11,595,325 | 616,491 | 616,491 | 616,491 | 616,491 |

Table 12: Number of inferring samples for the tasks in SMARTSPANNER (SSN) and SPANNER (SN) methods on the test datasets of CoNLL03, FEW-NERD and GENIA (1,000, 2,000, 5,000 and all sentences for training respectively).

| # Sents | Tasks | CoNLL03 | | | FEW-NERD | | | GENIA | | |
|---|---|---|---|---|---|---|---|---|---|---|
| | | $P$ | $R$ | $F1$ | $P$ | $R$ | $F1$ | $P$ | $R$ | $F1$ |
| 1,000 | NEH | 82.87 | 95.13 | 88.52 | 72.45 | 91.88 | 81.01 | 66.58 | 88.62 | 76.02 |
| | SP | 70.65 | 69.25 | 69.94 | 48.45 | 12.18 | 19.46 | 28.52 | 69.57 | 40.45 |
| | NEH + SP | 77.85 | 67.75 | 72.45 ($\uparrow$ **2.51**) | 49.74 | 12.18 | 19.57 ($\uparrow$ **0.11**) | 41.36 | 65.46 | 50.66 ($\uparrow$ **6.21**) |
| 2,000 | NEH | 91.81 | 97.01 | 94.34 | 78.94 | 93.35 | 85.54 | 72.20 | 90.39 | 80.27 |
| | SP | 74.00 | 81.40 | 77.52 | 45.81 | 27.16 | 34.07 | 37.28 | 82.04 | 51.26 |
| | NEH + SP | 87.16 | 80.66 | 83.78 ($\uparrow$ **6.26**) | 48.54 | 27.02 | 34.69 ($\uparrow$ **0.62**) | 54.50 | 77.35 | 63.94 ($\uparrow$ **12.68**) |
| 5,000 | NEH | 95.56 | 97.66 | 96.60 | 82.89 | 93.96 | 88.08 | 78.56 | 90.20 | 83.98 |
| | SP | 82.23 | 90.46 | 86.15 | 52.50 | 59.40 | 55.74 | 49.14 | 86.72 | 62.74 |
| | NEH + SP | 89.98 | 89.88 | 89.93 ($\uparrow$ **3.78**) | 57.97 | 58.56 | 58.26 ($\uparrow$ **2.52**) | 65.67 | 81.20 | 72.61 ($\uparrow$ **9.87**) |
| ALL | NEH | 96.42 | 97.95 | 97.18 | 87.31 | 94.56 | 90.79 | 81.71 | 89.64 | 85.49 |
| | SP | 89.44 | 91.94 | 90.68 | 61.74 | 71.55 | 66.28 | 55.92 | 88.36 | 68.47 |
| | NEH + SP | 91.25 | 91.56 | 91.40 ($\uparrow$ **0.72**) | 67.45 | 70.23 | 68.81 ($\uparrow$ **2.53**) | 71.61 | 82.22 | 76.54 ($\uparrow$ **8.07**) |

Table 13: The average values of precision ($P$), recall ($R$) and $F1$ scores of the two tasks (NEH and SP) in SMARTSPANNER on the test datasets of CoNLL03, FEW-NERD and GENIA in 10 independent runs (1,000, 2,000, 5,000 and all sentences for training respectively).