# OpenReview forum: "SmartSpanNER: Making SpanNER Robust in Low Resource Scenarios"
_EMNLP/2023/Conference — EMNLP 2023 Findings_

### Official Review · Reviewer_o3MJ · 2023-08-01

**Soundness:** 3

**Excitement:**

4: Strong: This paper deepens the understanding of some phenomenon or lowers the barriers to an existing research direction.

**Missing References:**

- The following paper may address a similar setting: Weng and Zhang, Named Entity Recognition Based on BERT-BiLSTM-SPAN in Low Resource Scenarios

**Paper Topic And Main Contributions:**

This paper points out that span-based named entity recognition (SpanNER) models do not perform well in low-resource settings with about 1000 sentences of training data, and proposes SmartSpanNER, which is based on multi-task learning with two tasks: one to predict the named entity head (NEH) and the other to classify spans, to address this problem. Experiments using four representative datasets, CoNLL03 and Few-NERD as flat NER data, and GENIA and ACE0 as nested NER data, show that SpanNER's performance degrades significantly when training data is small, and SmartSpanNER performs relatively robustly even when training data is small. In addition, a comparison of training and inference times between SpanNER and SmartSpanNER was also conducted, showing that SmartSpanNER can be trained and inferred in a relatively short time because the introduction of NEH narrows down the number of candidates.

**Reasons To Accept:**

- The problem that motivates the study and the model for solving the problem are explained clearly.
- This paper shows experimentally that span-based NER models do not perform robustly when the training data size is small.
- This paper proposes SmartSpanNER, which models NER as a multitask learning problem with two tasks: named entity head (NEH) prediction and span classification and shows through experiments that it performs relatively robustly even with small training data.
- After discussing the efficiency of learning and inference between SpanNER and SmartSpanNER, the paper shows through experiments that the latter is capable of learning and inference in a shorter period of time.

**Reasons To Reject:**

- There does not seem to be a major problem, but a minor problem is that the development set is not used to determine the two hyperparameters, w and sp, and they seem to be determined based on an analysis using a test set.

**Reproducibility:**

4: Could mostly reproduce the results, but there may be some variation because of sample variance or minor variations in their interpretation of the protocol or method.

**Reviewer Confidence:**

3: Pretty sure, but there's a chance I missed something. Although I have a good feel for this area in general, I did not carefully check the paper's details, e.g., the math, experimental design, or novelty.

**Typos Grammar Style And Presentation Improvements:**

- Line 024: information extract task -> information extraction task
- The font in Tables 4, 5, 10, and 11 is too small.

---

> ### Author Rebuttal · Authors · 2023-08-26
>
> Thanks for your careful and valuable comments. We will explain your concerns point by point.
>
> *Q1: There does not seem to be a major problem, but a minor problem is that the development set is not used to determine the two hyperparameters, w and sp, and they seem to be determined based on an analysis using a test set.*
>
> A1: We fully agree with this point and will consider making improvements to the experiment.
>
> *Q2: Missing References: The following paper may address a similar setting: Weng and Zhang, Named Entity Recognition Based on BERT-BiLSTM-SPAN in Low Resource Scenarios*
>
> A2: Thank you for providing the paper. We will cite it in the revision.
>
> *Q3: Typos Grammar Style And Presentation Improvements: Line 024: information extract task -> information extraction task; The font in Tables 4, 5, 10, and 11 is too small.*
>
> A3: Thank you for pointing out these issues. We will fix them in the revision.
>
> Thanks again for your insightful comments, which are very instrumental in improving the quality of our paper.

---

### Official Review · Reviewer_f3si · 2023-08-04

**Soundness:** 3

**Excitement:**

3: Ambivalent: It has merits (e.g., it reports state-of-the-art results, the idea is nice), but there are key weaknesses (e.g., it describes incremental work), and it can significantly benefit from another round of revision. However, I won't object to accepting it if my co-reviewers champion it.

**Missing References:**

Named Entity Recognition as Structured Span Prediction(https://aclanthology.org/2022.umios-1.1) (Zaratiana et al., UM-IoS 2022)

**Paper Topic And Main Contributions:**

This paper introduces SmartSpanner, a novel approach to enhance the robustness of Spanner i.e span-level prediction method for Named Entity Recognition. SmartSpanner incorporates a Named Entity Head (NEH) prediction task alongside span classification to address standard Spanner shortcomings: reduce sensitivity to training data size and to reduce the number of candidate span both during training and inference. The author conducted experiment on diverse range of datasets of flat and nester NER to validate their hypothesis.

**Questions For The Authors:**

* The max span width K is set to 20 in the paper which is arbitrary. I think one should check for the max entity width in the training set to decide this hyperparameter. A lower size of K should also increase the speed of the baseline Spanner. An analysis of the effect of K on speed and performance can quantify this issue.

* Why is NEH and Span classification have 1 and 2 layers respectively? What is the rationale for this decision?

* Suggestion for improving the baseline: training the model for more steps; tune the learning rate and the weight decay; etc.

**Reasons To Accept:**

* The idea is interesting as it prevent from enumerating from large number of candidates in span-based NER.
* The model is apparently more data-efficient than the standard span-based NER.
* The paper contains detailed analysis..

**Reasons To Reject:**

* While the authors claimed that the standard Spanner is sensitive to training data size, it seems like the model is rather poorly tuned. For instance, Zaratiana et al., (2022) use the standard Spanner on CoNLL-2003 with 1000 data obtained around 77 F1 (Table 6), while this paper reports an F1 score of 17.28. I suggest the authors review their baselines. Consequently, the authors claim that Sequence Labeling models are more data-efficient than Spanner, which is not supported by evidence.

* The authors omit crucial details about decoding: as the Spanner uses a sort of local span classification, a structured decoding should be applied to ensure well-formedness of the output. For instance, for flat NER, there should be no span overlapping; and for Nested NER, completely nested spans should be allowed but no clashing.

* I understand that comparison to SOTA is not required, but the results on Nested NER are very low compared to existing models, demonstrating again that the models are not well-tuned.

**Reproducibility:**

3: Could reproduce the results with some difficulty. The settings of parameters are underspecified or subjectively determined; the training/evaluation data are not widely available.

**Reviewer Confidence:**

4: Quite sure. I tried to check the important points carefully. It's unlikely, though conceivable, that I missed something that should affect my ratings.

---

> ### Author Rebuttal · Authors · 2023-08-25
>
> Thanks for your careful and valuable comments. We will explain your concerns point by point.
>
> *Q1: The max span width K is set to 20 in the paper which is arbitrary. I think one should check for the max entity width in the training set to decide this hyperparameter. A lower size of K should also increase the speed of the baseline Spanner. An analysis of the effect of K on speed and performance can quantify this issue.*
>
> A1: Thumbs up for your attention to experimental details!  The reason for setting this parameter to 20 is to ensure that the proportions of spans with a length exceeding 20 in the four datasets (train, dev and test) do not exceed 1%, which are shown as follows.
>
> |Dataset | Train | Dev | Test|
> |  ----  | ----  |----  |----  |
> |CoNLL03| 0.000% | 0.000% | 0.000% |
> |FEW-NERD| 0.004% | 0.002% | 0.004%|
> |GENIA| 0.000% | 0.000% | 0.000% |
> |ACE05| 0.312% | 0.124%| 0.330%|
>
> We apologize for not clarifying this in the paper and will provide an explanation in the revision.
>
> We quite agree that a lower size of K could increase the speed of the baseline Spanner. Additionally, a smaller K size will also enhance the speed of SmartSpanNER. We will analyze the effect of K on speed and performance in the revision.
>
> *Q2: Why is NEH and Span classification have 1 and 2 layers respectively? What is the rationale for this decision?*
>
> A2: We greatly appreciate your attention to the details of the model. As mentioned in the paper, our implementation is based on the code (https://github.com/neulab/SpanNER) provided by Fu et al. (2021), where a two-layer network is used for Span classification. Therefore, we have also adopted this setting. Considering that NEH classification is less challenging compared to Span classification (which is also validated in our experiments), we have used a single-layer network for NEH classification. Additionally, we adopted the greedy decoding method provided by Fu et al. (2021) for the flat NER task. We apologize for not explaining this in the paper.
>
> *Q3: Suggestion for improving the baseline: training the model for more steps; tune the learning rate and the weight decay; etc.*
>
> A3: Great suggestion! We will definitely follow it to improve the baseline performance.
>
> *Q4: Missing References: Named Entity Recognition as Structured Span Prediction(https://aclanthology.org/2022.umios-1.1) (Zaratiana et al., UM-IoS 2022)*
>
> A4: Very practical paper, we will cite it in the revision.
>
> Thanks again for your insightful comments, which are very instrumental in improving the quality of our paper.

---

### Official Review · Reviewer_Y6rd · 2023-08-04

**Soundness:** 4

**Excitement:**

4: Strong: This paper deepens the understanding of some phenomenon or lowers the barriers to an existing research direction.

**Paper Topic And Main Contributions:**

This paper studies the named entity recognition task of both flat and nested entities. The paper contrasts the two formulations, standard sequence labeling, and the span-level prediction. They highlight the challenges of span-level prediction in low-resource settings and present a joint learning framework that significantly improves the system's performance.

They introduce an auxiliary task of named entity head prediction that identifies words that constitute the start of any NE span. They jointly train this with the standard NE span prediction and report results on four datasets, CoNLL03 and Few-NERD (flat), GENIA, and ACE05 (flat+nested). They also propose a negative sampling strategy that mines hard negatives based on the entity head.

They present strong results on all four datasets, especially in low-resource scenarios.


**Questions For The Authors:**

1. Shibuya and Hovy 2020 present better results than the ones found in this paper. 84.3 vs 82.0 on ACE05 and 77.4 vs 76.5 on GENIA. I couldn’t find any comparison in the current draft. Any reason why this comparison was not made in the current paper?
2. I see two sets of negative samples,  hard negatives that share the start token with an entity and soft negatives that are randomly sampled using `sp`. It would be useful to perform the following ablation. This will tease apart the effectiveness of each component.
        - SpanNER w/ original negative samples
        - SpanNER w/ hard negatives
        - SpanNER w/ hard + soft negatives
        - SpanNER w/ hard + soft negatives, joint learning (aka SmartSpanNER)
3. Additional empirical analysis that reports differences between nested and flat entities in these benchmarks, maybe at varying depths (1st, 2nd, 3rd, and 4th levels; see Table 1 in Shibuya and Hovy, 2020).
4. In Table 4, is the training time in seconds?
5. Why not just use h_i instead of s(i,i) in NEH (equation 5)?
6. I appreciate average scores over multiple runs, but it would be great if you can include the standard deviations in Table 3.

**Reasons To Accept:**

1. The choice of auxiliary task is well-motivated and shows strong results across all datasets. As highlighted by the results, the auxiliary task is simpler than the NER task itself and doesn't adversely affect the core NER task. It also provides signficant gains at inference time.
2. The negative sampling strategy includes hard and soft negatives. Hard negatives constitute non-entity spans that share the head, and soft negatives are randomly sampled. This strategy provides clear gains in training time.
3. The analysis section is quite thorough. Paper reports average results across 10 independent runs across all experiments, studies the impact across varying training sizes, and sampling probabilities.

**Reasons To Reject:**

Some weaknesses include,

1. The paper motivates span-based methods on their ability to deal with nested entities. However, it doesn't analyze the effectiveness of the proposed method on nested entities from GENIA and ACE05 datasets. It doesn't compare against the state-of-the-art work (Shibuya and Hovy, 2020) on these two datasets. Interestingly, this paper underperforms the results from Shibuya and Hovy, 2020.
2. The two core contributions of this paper, the auxiliary task, and the sampling strategy seem quite effective. But it is hard to tease apart the effect of each. Some additional ablations that compare the effectiveness of hard, soft negatives, and joint learning can improve the claims of this paper.
3. A minor point about the choice of datasets in this paper. All the datasets in the paper contain large amounts of training data. It would be interesting to see the effectiveness of the proposed method in true low-resource settings. This could include NER on specialized domains, maybe? (see Zhang et al., 2021: https://academic.oup.com/jamia/article/28/9/1892/6307885)).

**Reproducibility:**

4: Could mostly reproduce the results, but there may be some variation because of sample variance or minor variations in their interpretation of the protocol or method.

**Reviewer Confidence:**

4: Quite sure. I tried to check the important points carefully. It's unlikely, though conceivable, that I missed something that should affect my ratings.

---

> ### Author Rebuttal · Authors · 2023-08-25
>
> Thanks for your careful and valuable comments. We will explain your concerns point by point.
>
> *Q1. Shibuya and Hovy 2020 present better results than the ones found in this paper. 84.3 vs 82.0 on ACE05 and 77.4 vs 76.5 on GENIA. I couldn’t find any comparison in the current draft. Any reason why this comparison was not made in the current paper?*
>
> A1: The aim of this paper is to improve the performance of SpanNER in low-resource scenarios by introducing the auxiliary task (NEH prediction). Therefore, we focused on comparing it with the SpanNER method. We will add comparisons with Shibuya and Hovy 2020 in the revision.
>
> *Q2. I see two sets of negative samples, hard negatives that share the start token with an entity and soft negatives that are randomly sampled using sp. It would be useful to perform the following ablation. This will tease apart the effectiveness of each component. - SpanNER w/ original negative samples - SpanNER w/ hard negatives - SpanNER w/ hard + soft negatives - SpanNER w/ hard + soft negatives, joint learning (aka SmartSpanNER)*
>
> A2: Good idea!  We will add this ablation experiment in the revision.
>
> *Q3. Additional empirical analysis that reports differences between nested and flat entities in these benchmarks, maybe at varying depths (1st, 2nd, 3rd, and 4th levels; see Table 1 in Shibuya and Hovy, 2020).*
>
> A3: Thank you for pointing out this! We will report the analysis in the revision.
>
> *Q4. In Table 4, is the training time in seconds?*
>
> A4: Yes, we have indicated it in the description of Table 4.
>
> *Q5. Why not just use h_i instead of s(i,i) in NEH (equation 5)?*
>
> A5: Because we treat NEH as a special span, we represent it this way. In fact, we can use h_i to represent s(i,i) in equation 5.
>
> *Q6. I appreciate average scores over multiple runs, but it would be great if you can include the standard deviations in Table 3.*
>
> A6: Thank you very much for your appreciation! We will include the standard deviations in the revision.
>
> *Q7:  It would be interesting to see the effectiveness of the proposed method in true low-resource settings. This could include NER on specialized domains, maybe? (see Zhang et al., 2021: https://academic.oup.com/jamia/article/28/9/1892/6307885)).*
>
> A7: Your suggestion is quite valuable, and we will consider validating our method in the true low-resource settings.
>
> Thanks again for your insightful comments, which are very instrumental in improving the quality of our paper.

---

### Meta-Review · Area_Chair_rfwc · 2023-09-16

**Recommendation:** 4

**Metareview:**

This paper proposes an auxiliary task for span-level prediction for NER to improve efficiency/robustness. They provide a thorough evaluation on multiple benchmarks, showing the effectiveness of their approach. Although in their original submission, their baselines were extremely low (17 vs 77SOTA); this has been resolved in the rebuttal. One of the reviewers still worries about the authors using more hyperparameter tuning for the proposed method compared to the baseline (private discussion); but from section 4.2 it seems like they were tuned the same way (and the rebuttal only adds the number of epochs). Therefore, the results seems sound (with a much better baseline, trends remain). In conclusion: a simple and motivated approach is introduces, which improves performance/robustness, this seems a valuable contribution.

Remaining weaknesses as reported by the reviewers are: no real low-resource setting is tested, no separate evaluation on nested entities, no ablation of the sampling strategy that was also introduced (besides the auxiliary task), tuning of 2 hyperparameters on test data. Furthermore, I think a (qualitative) analysis of which cases have improved could be expected from a long paper.

---

### Decision · Program_Chairs · 2023-10-07

**Decision:**

Accept-Findings

**Comment:**

This paper proposes an auxiliary task for span-level prediction for NER to improve efficiency/robustness. They provide a thorough evaluation on multiple benchmarks, showing the effectiveness of their approach. Although in their original submission, their baselines were extremely low (17 vs 77SOTA); this has been resolved in the rebuttal. One of the reviewers still worries about the authors using more hyperparameter tuning for the proposed method compared to the baseline (private discussion); but from section 4.2 it seems like they were tuned the same way (and the rebuttal only adds the number of epochs). Therefore, the results seems sound (with a much better baseline, trends remain). In conclusion: a simple and motivated approach is introduces, which improves performance/robustness, this seems a valuable contribution.

Remaining weaknesses as reported by the reviewers are: no real low-resource setting is tested, no separate evaluation on nested entities, no ablation of the sampling strategy that was also introduced (besides the auxiliary task), tuning of 2 hyperparameters on test data. Furthermore, I think a (qualitative) analysis of which cases have improved could be expected from a long paper.